# Sustainable Intensification of Rice Fallows with Oilseeds and Pulses: Effects on Soil Aggregation, Organic Carbon Dynamics, and Crop Productivity in Eastern Indo-Gangetic Plains

Kirti Saurabh [1], Rakesh Kumar [1,*], Janki Sharan Mishra [2], Anil Kumar Singh [1], Surajit Mondal [1], Ram Swaroop Meena [3], Jaipal Singh Choudhary [4], Ashis Kumar Biswas [5], Manoj Kumar [1], Himadri Shekhar Roy [6], Nongmaithem Raju Singh [7], Sushil Kumar Yadav [8], Ashutosh Upadhyaya [1], Hansraj Hans [1], Pawan Jeet [1], Prem Kumar Sundaram [1] and Rohan Kumar Raman [1]

1   ICAR-Research Complex for Eastern Region, Patna 800 014, India
2   ICAR-Directorate of Weed Research, Maharajpur, Jabalpur 482 004, India
3   Department of Agronomy, Institute of Agricultural Sciences, Banaras Hindu University, Varanasi 221 005, India
4   ICAR-RCER, Farming System Research Centre for Hill and Plateau Region, Plandu, Ranchi 834 010, India
5   ICAR-Indian Institutes of Soil Science, Bhopal 462 038, India
6   ICAR-Indian Agricultural Statistics Research Institute, New Delhi 110 012, India
7   ICAR-Research Complex for North Eastern Hill Region, Umiam 793 103, India
8   Natural Resource Management (Soil Science and Agricultural Chemistry), Bihar Agricultural University, Sabour, Bhagalpur 813 210, India
*   Correspondence: rakesh.kumar22@icar.gov.in; Tel.: +91-8257910434

**Abstract:** Climate-smart agriculture (CSA) practices are becoming increasingly important due to their better adaptability to harsh climatic conditions (in general) and the unpredictability of monsoons in India (in particular). Conventional rice cultivation (e.g., PTR) involves intensive tilling followed by intensive puddling in standing water that destroys the soil aggregation and depletes carbon pools. Therefore, alternative crop establishment methods need to be devised for the sustainability of system productivity, and the suitabilities of potential oilseeds and pulses need to be tested for cropping intensification in rice-fallow regions. Hence, an ongoing experiment (implemented in 2016) was evaluated to identify the appropriate CSA management practices in restoring soil C and physical health under diversified cropping systems in the rice-fallow system of eastern India. Six tillage and crop establishment methods along with residue management were kept as the main plots [zero-till-direct-seeded rice (ZTDSR), conventional-till-DSR (CT-DSR), puddled transplanted rice (PTR), ZTDSR with rice residue retentions (ZTDSR$_{R+}$), CTDSR with rice residue retention (CTDSR$_{R+}$), PTR with rice residue retention (PTR$_{R+}$)] while five winter/post-rainy crops (oilseeds and pulses) were raised in a subplot. In the ZTDSR$_{R+}$ production system, soil macro-aggregate (%), macro-aggregate-associated C, MWD, and GMD of aggregates increased by 60.1, 71.3, 42.1, and 17.1%, respectively, in comparison to conventional tillage practices (PTR). The carbon management index (CMI) was 58% more in the ZTDSR$_{R+}$ production system compared to PTR. Among the winter crops, chickpeas recorded higher values of soil structural indices and C content. In the PTR production system, system productivity, in terms of rice equivalent yield, was comparable to ZTDSR$_{R+}$. ZT with residue retention in rice followed by post-rainy/winter pulses led to higher C content and structural stability of the soil. Thus, CSA management practices can improve the crop productivity as well as soil health of rice-fallow production systems of eastern India and comparable agroecotypes of South Asia.

**Keywords:** climate-smart agriculture (CSA); oilseeds; pulses; residue retention; rice fallow; soil health; zero-tillage

## 1. Introduction

Climate-resilient and sustainable agricultural production systems are important for maintaining the equilibrium between higher productivity and resource scarcity [1]. The situation is more critical for rice-based production systems, which support the livelihoods of 300 million people and cover ~140 M ha in Asia [2]. It is argued that synergistic crop production with natural resources is possible and this can be accomplished by increasing resource-use efficiency (RUE), input substitution, and designing sustainable climate-resilient cropping systems [3]. In many parts of the eastern Indo-Gangetic Plains (EIGP) of India, the farmers are forced to keep the fields 'fallow' after rice harvesting due to several abiotic, biotic, and socioeconomic factors [4]. Low fertility, problematic soils, unavailability of assured irrigation facilities, variable environmental conditions, and poor socioeconomic conditions are some of the reasons for fallowing after rice harvesting in the regions [4]. During the wet season, anaerobic conditions in rice and puddling destroy the soil aggregation and organic carbon (C) content of the soil, causing lower productivity of succeeding winter/post-rainy crops [5]. Moreover, anaerobic soil conditions deteriorate the favorable microbial balance of the rhizosphere. Lack of scientific information, poor availability of the improved seeds, insufficient technical supervision, seed storage, irrigation, and marketing further contribute to the acreage of rice fallows [5]. About 11.7 M ha area of India remains fallow after rice harvesting and it constitutes ~79% of total rice fallow areas of South Asia (15 M ha) [4].

Although India is approaching self-sufficiency in pulses, the import of oilseeds is still a major concern. In India, oilseeds are cultivated over an area of 25 M ha. At present (2020–2021), the total production of oilseeds in the country is 36.1 M tons [6]. Assuming a country-wide average of 28% oil recovery [7], 36 million tons of oilseeds will yield ~10 million tons of edible oil. The total edible oil requirement in India in 2022 is estimated to be 33.2 million tons assuming a per capita consumption of 22 kg per annum [6]. In 2019–2020, India imported a total of 13.35 million tons of vegetable oils costing INR 61,559 crores. To achieve oilseed self-sufficiency in this country, the productivity of oilseeds needs to be doubled via improved varieties and crop management practices, and also by increasing the area under oilseed cultivation. Therefore, the production of oilseeds needs to be promoted in new niche areas, such as rice fallow unutilized land with proper strategic planning, as well as input and policy support. If at least 50% of rice-fallow areas (~6.0 M ha) are brought under the oilseed production with average productivity of 500 kg ha$^{-1}$, ~3 million tons of the oilseeds can be added to the national oilseed basket in the country [8].

Numerous physical and chemical processes in the soil are mediated by the soil aggregate and SOC, including nutrient recycling, soil hardening, soil loss owing to erosion, root penetration, and crop development [9,10]. In rice-growing regions, puddling (wet tillage) has adverse impacts on soil aggregations, beneficial microbial activity, and complete soil conditions [11]. Other alternative methods, in addition to soil and crop management, are critical to reduce the negative effects of conventional TPR production systems and ensuring long-term sustainability [12]. Improved soil structure and aggregate stability are widely used as markers of soil conditions as they are essential for greater soil fertility, long-term stability, and crop production [13]. Thus, enhancing the soil carbon content with the proper agricultural soil management practices can mitigate climate change, food and nutritional insecurity in the regions.

Several CSA technologies can improve crop productivity and build resilience to the climatic changing risks for smallholder farmers of rice-fallow areas. This can be done by choosing short-duration high-yielding rice varieties that have fewer water requirements and are tolerant to abiotic stresses, direct-seeded rice crop establishment methods, retaining rice stubble for moisture conservation, and introducing short-duration varieties of oilseeds and pulses [14]. As ~15 M ha in South Asia remains fallow (uncultivated) after the rice harvest each year, there are great scopes to increasing the cropping intensity as well as overall system productivity, which will concurrently improve profitability and can be



the instrument used to achieve food and nutritional security in the regions. To achieve self-sufficiency in oilseeds and pulses and to curtail the outflow of the foreign currency for import, short-duration high-yielding oilseeds (e.g., mustard, linseed, and safflower) and pulses (viz. chickpea, lentil, lathyrus, and field pea) may be grown in the rice-fallow system depending on the residual soil moisture and lifesaving irrigation [14]. Additionally, pulses facilitate soil health restoration by fixing atmospheric nitrogen (N) and adding crop biomass, which ultimately improves the soil organic C (SOC) status of the soil [14]. Moreover, the surface cover offered by post-rainy/winter crops after rice harvesting protects the soil from erosion by external forces, such as wind and water, and lowers the oxidation of soil organic matter (SOM). In this context, an experiment is being initiated with the hypothesis that the adoption of climate-smart agriculture (CSA) practices, such as alternative rice establishment methods other than puddling, the retention of crop residues for better soil moisture conservation, adopting zero-tillage (ZT) during post-rainy/winter crops for improving soil health and reducing greenhouse gas (GHG) emission, and introducing short-duration high–yielding pulses and oilseeds can augment the overall system productivity, profitability, and soil resilience of rice-fallow areas. The findings may be useful for devising suitable cropping systems for rice-fallow areas of eastern India and similar agroecotypes of the world.

## 2. Materials and Methods

### 2.1. Study Location

Experimentation is being carried out as part of an ongoing experiment (initiated in 2016) at the Sabajpura Research Farm (25°34′ N, 85°03′ E, and 51 m AMSL), the ICAR-Research Complex for Eastern Region, Patna, Bihar, India (Figure 1). The weather parameters are shown in Supplementary Figure S1. The soil is silty clay loam in texture (silt: 53.3%, clay: 36.0%, and sand: 10.7%) with a pH of 7.58; Walkley-Black C:5.6 g kg$^{-1}$; available N, P, and K: 183, 51, and 250 kg ha$^{-1}$, respectively; bulk density: 1.63 Mg m$^{-3}$ and DTPA (diethylenetriaminepentaacetic acid) extract. Fe, Zn, Mn, and Cu are 71.3, 0.74, 12.4, and 3.54 ppm, respectively.

### 2.2. Field Management and Experimental Design

The field experiment was established during the rainy season of 2016 in a split-plot design with three replications. Six tillage and crop-establishment methods along with crop residue management were kept as the main plot (zero-till-direct-seeded rice (ZTDSR), conventional-till-DSR (CT-DSR), puddled transplanted rice (PTR), ZTDSR with rice residue retention (ZTDSR$_{R+}$), CTDSR with rice residue retention (CTDSR$_{R+}$), and PTR with rice residue retention (PTR$_{R+}$), while five post-rainy/winter crops (oilseeds i.e., safflower, linseed, mustard; pulses i.e., chickpea, lentil) were raised in sub-plots. During the rainy season (June–October), rice seeds (cv. Swarna Shreya) were sown in both plots (for DSR at 30 kg ha$^{-1}$) and nursery beds (for transplanted rice at 20 kg ha$^{-1}$) during the third week of June. Later, in the first week of July, 21-day-old rice seedlings were transplanted. N: P: K fertilizer doses were 120–60–40 kg ha$^{-1}$ for rice. At transplanting, one-third of nitrogen, as well as entire doses of phosphorus and potassium, were applied. At the maximum tillering and panicle initiation stages, the rest of the nitrogen was used in equal halves in the form of urea. Following the harvesting of rice, five winter/post-rainy crops *viz.* chickpeas, lentils, safflowers, linseeds, and mustard were sown in the fourth week of October. Chickpea seeds (cv. Pusa 256) at 80 kg ha$^{-1}$ were sown with 20–50 (NP) fertilizer doses. Lentils (cv. HUL 57) at 40 kg ha$^{-1}$ were sown after rice with 20–50 (NP) fertilizer doses. In oilseed crops, linseeds (cv. T97) at 25 kg ha$^{-1}$ were sown with 50–30–20 (NPK) fertilizer requirements. Mustard (cv. Proagro 5111) at 5 kg ha$^{-1}$ was sown with 40–20–20 (NPK) fertilizer requirements. Treatment descriptions are given in Supplementary Table S1.

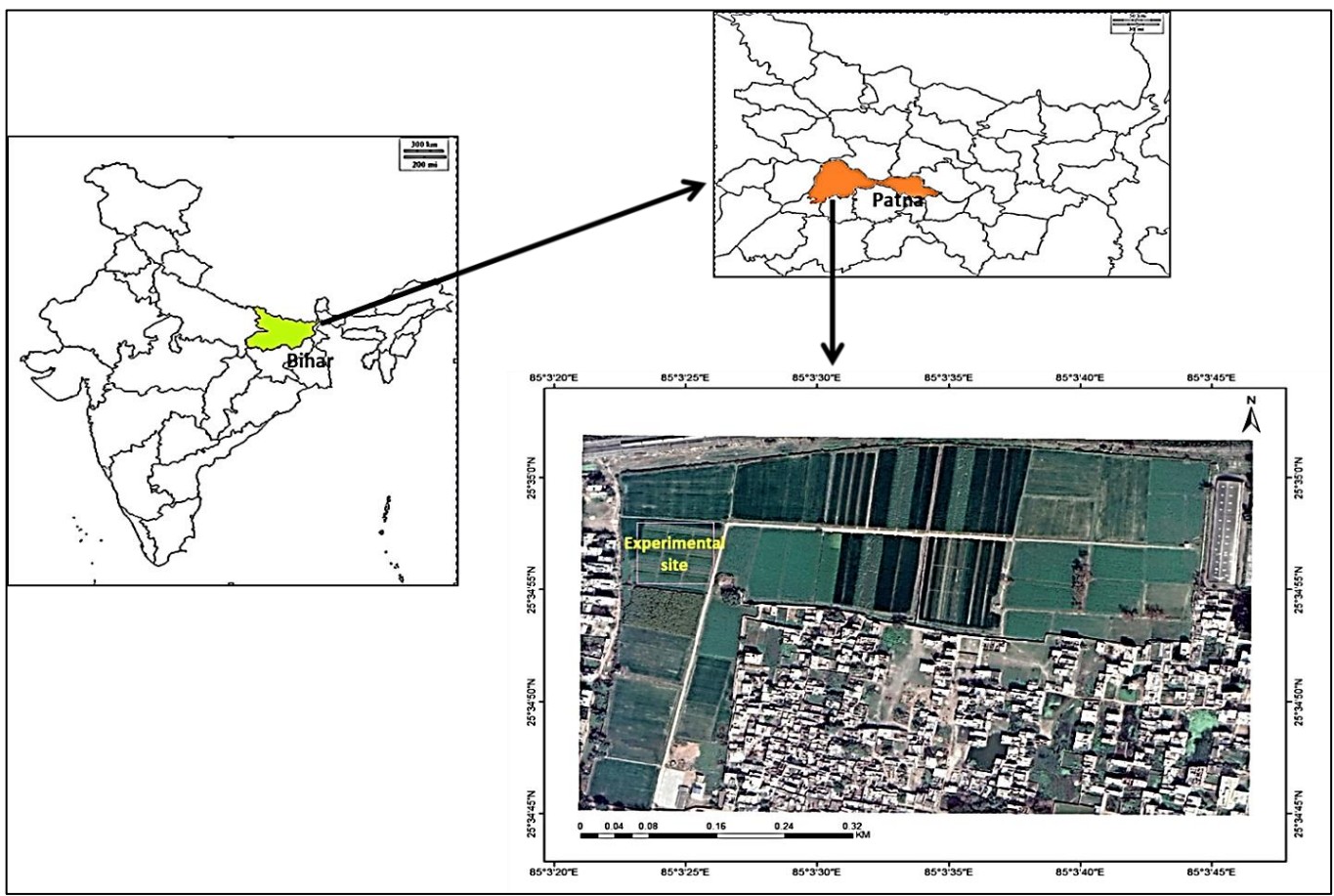

**Figure 1.** The location of the experimental site.

*2.3. Analysis of Soil Samples*

We took the soil samples after 4-years of experimentation (2019) using a soil core sampler, from soil depths of 0–15 and 15–30 cm. Samples were analyzed for SOC concentrations [15], improved chromic acid digestion techniques for total organic C [16], the modified Walkley and Black approach for carbon fractions at the various degrees of oxidation [17], available N (Kjeldahl method), available P analyzed by the Olsen method [18], and available K by the Hanway and Heidel method [19]. Air-dried unground samples were sieved at 5 mm and utilized to estimate the aggregate size distributions using the wet sieving technique [20]. Dry soil aggregates were sieved at 0.15 mm to analyze the total organic C (TOC). Chemicals and solvents used in the analysis were obtained from Merck and Himedia. All assays were measured in triplicate for each treatment. Mean weight diameter of aggregates (MWD), geometric mean diameter of aggregates (GMD), unstable aggregate index ($E_{LT}$), and fractal dimension (D) parameters were calculated to determine the aggregation statuses of the soils. Different sizes of the soil aggregates were separated by the wet sieving method and different aggregate indices were calculated by the following formulas [21].

$$\text{MWD (mm)} = \sum_{i=1}^{n}(Xi\ Wi)\ /\ \sum_{i=1}^{n} Wi \tag{1}$$

$$\text{GMD (mm)} = \exp((\sum_{i=1}^{n} Wi\log Xi)\ /\ (\sum_{i=1}^{n} Wi)) \tag{2}$$

where *Wi* denotes the aggregate retained across the sieve (g) and Xi denotes the size class mean diameter (mm).

$$E_{LT} = \frac{W_T - W_{0.25}}{W_T} \times 100\% \tag{3}$$

where $W_T$ denotes the total weight of the soil and the weight of the water-stable aggregate is denoted by $W_{0.25}$.

The fractal dimension ($D$) was estimated from the following equation [22]:

$$\frac{M(r < \overline{x}_i)}{M_T} = \left( \frac{\overline{x}_i}{x_{max}} \right)^{3-D} \tag{4}$$

Taking logarithms of the formula above (4):

$$log\left[ \frac{M(r < \overline{x}_i)}{M_T} \right] = (3 - D)log\left( \frac{\overline{x}_i}{x_{max}} \right) \tag{5}$$

$D$ can be obtained from Formulae (4) and (5) by using data fitting.

The weight of the soil aggregate of a particular size is given by xi in the formulae, the weight of the aggregate with a diameter less than $x_i$ is $M (r < x_i)$, and $x_{max}$ is the maximum diameter of the aggregate.

### 2.4. Pools of Oxidizable Organic Carbon

The quality of SOC was measured in terms of the degrees of oxidizability as very labile, labile, less labile, and non-labile pool by oxidation with 12 N, 18 N, and 24 N $H_2SO_4$ (acid /aqueous ratios of 0.5:1, 1:1, and 2:1, respectively) [17]. The quantity of C causes TOC to be partitioned into four separate organic C pools.

Fractions I (very labile C-VLC): organic C oxidizable at 12.0 N $H_2SO_4$.
Fractions II (labile C-LC): organic C oxidizable at 18.0 N–12.0 N $H_2SO_4$.
Fractions III (less labile C-LLC): organic C oxidizable at 24.0 N–18.0 N $H_2SO_4$.
Fractions IV (non-labile C-NLC): TOC-C oxidizable at 24.0 N $H_2SO_4$.
Active pool (AP) =VLC + LC.
Passive pool(s) (PP) = LLC + NLC.

The C management index (CMI) was calculated using the mathematical methodology given by Blair et al. [23].

$$\text{CMI} = \text{CPI} \times \text{LI} \times 100 \tag{6}$$

where the C pool index (CPI) = C pools in the samples (mg kg$^{-1}$ of soil)/C pools in the references (mg kg$^{-1}$ of soil),

Lability index (LI) = ((Fraction I/TOC) * 3 + (Fraction II/TOC) * 2 + (Fraction III/TOC) * 1

The C pool of the control plot is used as the C pool in the reference [23].

Aggregate associated total organic C was determined as described for bulk soil and the C preservation capacity (*CPC*) was calculated as:

$$CPC = \frac{WSACi * WSAi}{100} \tag{7}$$

where, *WSACi* is the TOC in water-stable aggregates of >2 mm, 2–0.5 mm, 0.5–0.25, 0.25–0.125, or 0.125–0.053 mm, and *WSAi* is the percentage of water stable aggregates of >2 mm, 2–0.5 mm, 0.5–0.25, 0.25–0.125, or 0.125–0.053 mm [24].

### 2.5. System Rice Equivalent Yield (SREY)/System Productivity

Rice equivalent yield (REY)/system productivity was obtained by changing the crop yields other than rice into REY based on the minimum support price (MSP) announced for each crop every year by the Government of India (GOI).

REY = Yx (Px/Pr),

where Yx denotes the yield of non-rice crops (kg ha$^{-1}$), Px denotes the price of non-rice crops (INR kg$^{-1}$), and Pr denotes the price of rice (INR kg$^{-1}$).

Rice yields from the rainy and winter seasons were added together and are represented as kg ha$^{-1}$ for system REY (SREY) calculations.

### 2.6. Statistical Analysis

Using the proper layout and framework of the split-plot design, we performed an analysis of variance and Tukey's test for the post hoc analysis by using SAS 9.3 [25]. The differences between the selected treatment means were analyzed using a mixed model approach. Here, replication and tillage (main plots) are assumed as random effects and the cropping pattern (sub-plot) is assumed as a fixed effect in the mixed modeling approach. Further, R version 4.2.1 [26] was used to generate the correlation plot. Data were tested for normality using the Q-Q plot as well as the Shapiro–Wilk test, as needed.

## 3. Results

### 3.1. Distribution Characteristics of Water-Stable Soil Aggregates

Throughout the soil depth, 2–0.5 mm and 0.5–0.25 mm aggregates predominated, accounting for 32.1–48.5% and 14.9–26.3% of the total aggregates at 0–15 cm and 26.5–31.7% and 24.2–30.9% at 15–30 cm in different CERM and cropping rotation (CR) treatments, respectively (Table 1). Zero-tillage with residue retention treatments (ZTDSR$_{R+}$) showed significantly greater percentages of macroaggregates in 0–15 cm of soil compared to other treatments. Similarly, at the surface soil in residue-retained treatments, small aggregates of 0.25–0.125 mm size class were higher as compared to residue removed treatment. For >2.0 mm and 2–0.5 mm size classes, ZTDSR$_{R+}$ and the rice–chickpea (R-C) treatment outperformed at 0–15 cm depths. Microaggregate (0.125–0.053 mm) content did not significantly differ with different CERM treatments; however, among CR treatments, significantly higher fractions were observed under rice–safflower (R-SF) followed by rice–lentil (R-L) and rice–linseed (R-Li) in the surface soil. At lower soil depths (15–30 cm), a higher aggregate in class >2.0 mm was observed in ZTDSR$_{R+}$ but was at par with CTDSR$_{R+}$. Among the post-rainy/winter crop rotations, rice–chickpea (R-C), rice–safflower (R-SF), and rice–lentil (R-L) production systems had significantly higher percentages of all aggregates at 15–30 cm. The total water-stable aggregate was 48.1% higher in ZTDSR$_{R+}$ compared to PTR production systems at 0–15 cm depth. Whereas, among the winter crops rice–chickpea (R-C), rice–safflower (R-SF), and rice–linseed (R-Li) had comparatively higher total water-stable aggregates than others.

**Table 1.** Effect of crop establishment-cum-residue management practices and cropping rotations on the distribution of aggregates in the soil profile.

| Treatments | Aggregate Size Class (mm) | | | | | | | | | |
|---|---|---|---|---|---|---|---|---|---|---|
| | 0–15 cm | | | | | 15–30 cm | | | | |
| | Macroaggregate (%) | | | Microaggregate (%) | | Macroaggregate (%) | | | Microaggregate (%) | |
| | >2.0 | 2–0.5 | 0.5–0.25 | 0.25–0.125 | 0.125–0.053 | >2.0 | 2–0.5 | 0.5–0.25 | 0.25–0.125 | 0.125–0.053 |
| Crop establishment-cum-residue management (CERM) | | | | | | | | | | |
| ZTDSR$_{R+}$ | 6.14 a | 48.01 a | 24.62 a | 9.62 b | 7.17 a | 3.16 a | 30.93 a | 30.96 a | 20.03 a | 8.16 ab |
| ZTDSR | 3.19 bc | 46.04 ab | 19.01 ab | 8.59 b | 6.82 a | 1.98 b | 29.07 a | 27.07 b | 12.18 bc | 5.75 b |
| CTDSR$_{R+}$ | 3.93 bc | 37.25 bc | 24.51 a | 12.24 a | 6.07 a | 3.01 a | 30.02 a | 26.93 b | 15.6 4b | 10.54 a |
| CTDSR | 2.52 c | 37.82 bc | 20.65 ab | 8.51 b | 5.78 a | 1.87 b | 29.29 a | 25.91 b | 14.02 bc | 8.06 ab |
| PTR$_{R+}$ | 4.59 ab | 34.94 c | 25.24 a | 10.35 ab | 6.00 a | 1.67 b | 28.69 a | 24.24 b | 13.99 bc | 7.19 b |
| PTR | 2.36 c | 32.14 c | 14.91 b | 9.11 b | 6.36 a | 1.38 b | 28.52 a | 24.43 b | 8.03 d | 5.48 b |
| Cropping rotations (CR) | | | | | | | | | | |
| R-C | 5.05 a | 46.91 a | 17.13 c | 7.26 b | 5.47 b | 2.29 ab | 31.78 a | 25.22 a | 15.17 ab | 8.01 a |
| R-L | 2.26 c | 35.93 c | 22.28 b | 12.31 a | 6.80 ab | 1.57 b | 26.55 b | 26.71 a | 16.36 a | 7.90 a |
| R-SF | 4.21 ab | 36.86 bc | 19.75 bc | 13.93 a | 7.15 a | 2.66 a | 28.94 ab | 26.71 a | 16.43 a | 6.90 a |
| R-Li | 4.09 ab | 42.36 ab | 21.98 b | 7.87 b | 6.71 ab | 2.40 a | 30.67 ab | 27.95 a | 9.41 c | 8.26 a |
| R-M | 3.34 bc | 34.77 c | 26.30 a | 7.32 b | 5.72 b | 1.96 ab | 29.16 ab | 26.36 a | 12.54 b | 6.59 a |

Means followed by the same letters in the columns do not differ significantly from each other by the Tukey HSD test at 5% probability level (*n* = 3); ZTDSR: zero-tillage direct seeded rice; CTDSR: conventional-till-direct seeded rice; PTR: puddle transplanted rice; R+: 30% residue retention, R-C: rice–chickpea; R-L: rice–lentil; R-SF: rice–safflower; R-Li: rice–linseed; R-M: rice–mustard.

### 3.2. Stability of the Water-Stable Soil Aggregate

The stability of the water-stable soil aggregate was determined using GMD, MWD, and $E_{LT}$, which varied depending on the soil depth and treatment (Table 2). For all treatments, GMD and MWD were reduced when the depth increased. At 0–15 cm depths, $ZTDSR_{R+}$ improved MWD and GMD of aggregates by 13% and 6%, respectively, over the conventional tillage system (PTR). Among the winter crops, rice–chickpea (R-C) rotations showed the highest MWD and GMD values followed by rice–linseed (R-Li) at 0–15 cm depths. Significantly higher values of $E_{LT}$ in the PTR production system were observed in comparison to $ZTDSR_{R+}$ and $CTDSR_{R+}$ at both soil layers. Irrespective of soil depths, no significant differences were observed among different CR treatments.

**Table 2.** Influence of CERM and CR on MWD, GMD, and $E_{LT}$ at 0–15 cm and 15–30 cm soil depths.

| Treatment | 0–15 cm | | | 15–30 cm | | |
|---|---|---|---|---|---|---|
| | MWD (mm) | GMD (mm) | $E_{LT}$ | MWD (mm) | GMD (mm) | $E_{LT}$ |
| Crop establishment-cum-residue management (CERM) | | | | | | |
| $ZTDSR_{R+}$ | 0.97 a | 0.84 a | 11.60 c | 0.71 b | 0.71 bc | 14.89 d |
| ZTDSR | 0.93 ab | 0.83 a | 23.15 b | 0.73 ab | 0.75 ab | 29.68 bc |
| $CTDSR_{R+}$ | 0.87 c | 0.81 ab | 22.05 bc | 0.72 ab | 0.70 c | 24.38 c |
| CTDSR | 0.87 c | 0.79 b | 30.49 b | 0.71 b | 0.72 bc | 28.89 bc |
| $PTR_{R+}$ | 0.88 bc | 0.79 b | 24.87 b | 0.72 ab | 0.73 bc | 31.39 ab |
| PTR | 0.86 c | 0.79 b | 41.46 a | 0.77 a | 0.77 a | 37.61 a |
| Cropping rotation(s) (CR) | | | | | | |
| R-C | 1.01 a | 0.87a | 23.64 a | 0.71 a | 0.72 bc | 28.13 a |
| R-L | 0.80 d | 0.77 c | 27.21 a | 0.67 b | 0.70 c | 28.79 a |
| R-SF | 0.89 bc | 0.79 cd | 25.23 a | 0.74 a | 0.73 b | 25.23 a |
| R-Li | 0.92 b | 0.82 b | 23.69 a | 0.76 a | 0.74 ab | 29.56 a |
| R-M | 0.86 c | 0.80 bc | 28.25 a | 0.76 a | 0.75 a | 27.34 a |

Means followed by the same letters in the columns do not differ significantly from each other by Tukey's HSD test at 5% probability level (*n* = 3). ZTDSR: zero-till-direct seeded rice; CTDSR: conventional-till-direct seeded rice; PTR: puddle transplanted rice; R+: 30% residue retention, R-C: rice–chickpea; R-L: rice–lentil; R-SF: rice–safflower; R-Li: rice–linseed; R-M: rice–mustard.

### 3.3. Aggregates Fractal Dimension

According to the statistical analysis, there was a substantial difference in D values in both soil depths between CERM treatments (Figure 2). The D value varied from 2.05 to 2.54 at 0–15 cm and from 2.08 to 2.64 at 15–30 cm depth under different CERM and CR treatments. Overall the D value increased with increasing soil depth. In the 0–15 cm depth, PTR (2.63) treatment had a significantly higher D value than other treatments. At the 15–30 cm depths, $ZTDSR_{R+}$ maintained a significantly lower D value. At both depths, PTR treatments produced higher D values than other treatments. Among the winter crops, the rice–safflower (R-SF) rotation had lower D values at both 0–15 cm (2.33) and 15–30 cm (2.42) depths, though the differences were non-significant among themselves.

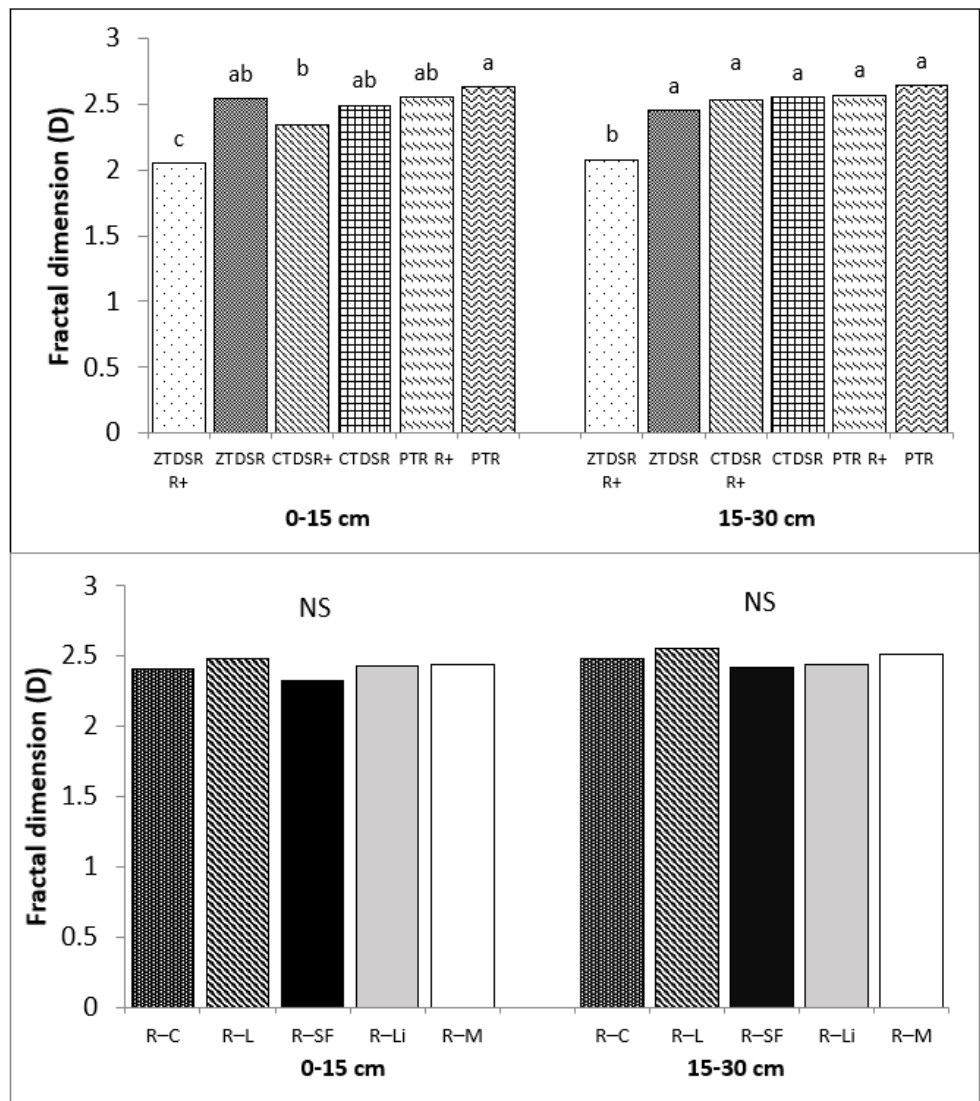

**Figure 2.** Fractal dimension (D) of the water aggregate as affected by the CERM and CR. ZTDSR: zero-till-direct seeded rice; CTDSR: conventional-till-direct seeded rice; PTR: puddle transplanted rice; R+: 30% residue retention, R-C: rice–chickpea; R-L: rice–lentil; R-SF: rice–safflower; R-Li: rice–linseed; R-M: rice–mustard. Means followed by the same letters do not differ significantly from each other by Tukey's HSD test at 5% probability level (*n* = 3).

### 3.4. Distribution of Water-Stable Aggregate-Associated Carbon

The CERM and CR treatments had substantial impacts on the TOC concentrations of aggregates. The TOCs associated with macro- and microaggregates reduced as soil depth increased (Table 3); topsoil had a larger aggregate-associated C content than underlying soil. ZTDSR$_{R+}$ production systems had the greatest TOCs in terms of macro-(>2 mm), meso-(2–0.5 mm and 0.5–0.25 mm), and micro-(0.25–0.125 mm) aggregates at upper soil depths. Whereas the lowest macro-and meso-aggregate-associated TOCs were in PTR production systems and the lowest coarse microaggregate-associated TOC was observed in CTDSR (5.58 g kg$^{-1}$ of soil) at 0–15 cm depths. In 15–30 cm of soil depth, a similar pattern was seen. Among winter crops, R-L, R-SF, and R-C rotations showed higher macro- and meso-aggregate-associated TOCs at 0–15 cm. The rice–safflower (R-SF) system had significantly higher coarse microaggregate-associated TOC (9.13 g kg$^{-1}$ of soil) in the 0–15 cm soil layer.

**Table 3.** Aggregate-associated carbon influenced by the CERM and CR at 0-15 cm and 15-30 cm soil depths.

| Treatments | Aggregate C (g kg$^{-1}$ Soil Aggregate) | | | | |
|---|---|---|---|---|---|
| | **CMacAC** | **MesoAC** | | **CMicAC** | **FMicAC** |
| **0–15 cm** | **>2 mm** | **2–0.5 mm** | **0.5–0.25 mm** | **0.25–0.125 mm** | **0.125–0.053 mm** |
| Crop establishment-cum-residue management (CERM) | | | | | |
| ZTDSR$_{R+}$ | 6.57 a | 9.16 a | 9.84 a | 9.31 a | 8.93 a |
| ZTDSR | 3.56 c | 7.51 bc | 7.70 bc | 7.88 bc | 6.85 bc |
| CTDSR$_{R+}$ | 4.58 b | 7.98 b | 8.52 b | 8.74 ab | 7.45 b |
| CTDSR | 2.06 d | 6.12 d | 6.63 c | 5.58 e | 6.17 cd |
| PTR$_{R+}$ | 4.20 bc | 6.62 cd | 7.08 c | 7.38 cd | 6.66 bcd |
| PTR | 2.27 d | 5.79 d | 4.73 d | 6.65 d | 5.79 d |
| Cropping rotations (CR) | | | | | |
| R-C | 3.78 b | 7.89 a | 7.79 a | 7.14 b | 8.31 a |
| R-L | 4.71 a | 7.56 ab | 7.68 a | 7.15 b | 6.20 c |
| R-SF | 4.01 b | 7.15 b | 7.94 a | 9.13 a | 7.01 b |
| R-Li | 3.05 c | 6.38 c | 7.45 a | 7.50 b | 6.77 bc |
| R-M | 3.84 b | 6.99 bc | 6.26 b | 7.03 b | 6.72 bc |
| **15–30 cm** | | | | | |
| Crop establishment-cum-residue management (CERM) | | | | | |
| ZTDSR$_{R+}$ | 5.76 a | 8.58 a | 7.72 a | 7.59 a | 7.37 a |
| ZTDSR | 3.63 bc | 6.14 bc | 6.17 bc | 6.47 abc | 6.09 abc |
| CTDSR$_{R+}$ | 5.92 a | 6.63 b | 7.07 ab | 6.84 ab | 6.77 ab |
| CTDSR | 3.00 c | 5.39 bc | 5.29 cd | 6.02 bc | 5.79 bc |
| PTR$_{R+}$ | 4.18 b | 6.57 b | 6.18 bc | 6.32 bc | 6.23 abc |
| PTR | 2.04 d | 5.08 c | 4.88 d | 5.44 c | 4.94 c |
| Cropping rotations (CR) | | | | | |
| R-C | 3.62 b | 6.89 a | 6.15 a | 6.63 a | 5.88 b |
| R-L | 4.05 ab | 6.75 a | 6.72 a | 6.71 a | 7.16 a |
| R-SF | 3.86 b | 6.23 a | 5.82 a | 6.44 a | 5.99 b |
| R-Li | 4.14 ab | 5.82 a | 5.97 a | 6.09 a | 5.75 b |
| R-M | 4.77 a | 6.30 a | 6.41 a | 6.36 a | 6.22 ab |

Means followed by the same letters in the columns do not differ significantly from each other by the Tukey HSD test at 5% probability level (*n* = 3). ZTDSR: zero-tillage direct seeded rice; CTDSR: conventional-till-direct seeded rice; PTR: puddle transplanted rice; R+: 30% residue retention, R-C: rice–chickpea; R-L: rice–lentil; R-SF: rice–safflower; R-Li: rice–linseed; R-M: rice–mustard. CMacAC—coarse macroaggregated carbon; MesoAC—meso-aggregated carbon. CmicAC—coarse micro-aggregated carbon; FmicAC—fine micro-aggregated carbon.

*3.5. Carbon Preservation Capacity (CPC) of the Different Aggregate Class*

In both the soil depths, macroaggregates were observed to retain a larger fraction of the TOC (Figure 3). As a result, depending on the treatment, the CPCs of the various aggregate sizes varied. Resource conservation methods affected the CPCs of aggregates of varied sizes. Under PTRR$_{R+}$ treatments, coarse microaggregates (0.25–0.125 mm) showed the highest capacity to capture the C. Residue retention also encouraged the higher CPCs in aggregates under upper soil depths. Higher C density in ZTDSR$_{R+}$ in meso-aggregates indicated that it had significance in soil C-sequestration. Oilseed and pulses-based cropping systems showed higher CPCs at both soil depths.

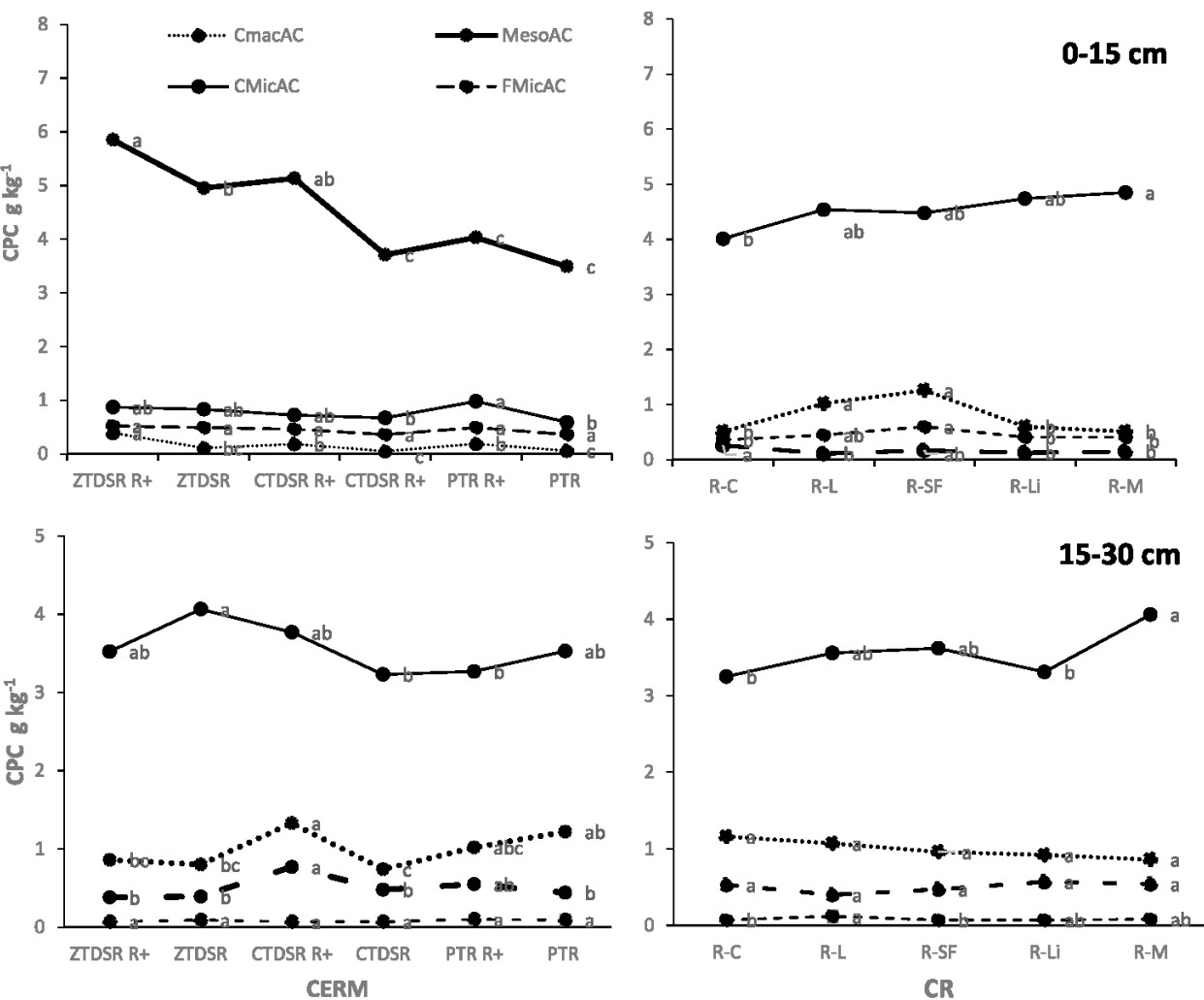

**Figure 3.** CPC (g kg⁻¹ of aggregate soil) of different soil aggregates at 0–15 cm and 15–30 cm as influenced by CERM and CR. ZTDSR: zero-tillage direct seeded rice; CTDSR: conventional-till-direct seeded rice; PTR: puddle transplanted rice; R+: 30% residue retention, R-C: rice–chickpea; R-L: rice–lentil; R-SF: rice–safflower; R-Li: rice–linseed; R-M: rice–mustard. CMacAC-coarse macroaggregated carbon; MesoAC—meso-aggregated carbon CMicAC–coarse micro-aggregated carbon; FMicAC–fine micro-aggregated carbon. Means followed by the same letters do not differ significantly from each other by Tukey's HSD test at 5% probability level (*n* = 3).

### 3.6. Fractions of the Bulk Soil Organic Carbon

The active C pool (AP) of the soil was significantly impacted due to various CERM and CR systems in all measurement depths (Figure 4). In general, AP was more in the upper soil depth than those in the 15–30 cm depth. In particular, soil in ZTDSR$_{R+}$ had a higher AP (7.23 g kg⁻¹ soil) and was trailed by the CTDSR$_{R+}$ production system (7.09 g kg⁻¹). Larger content of TOC was detected in ZTDSR, CTDSR, and PTR along with the residue integration. A significantly higher value of TOC was observed in R-SF, R-C, R-L, and R-M rotations at 0–15 cm depths.

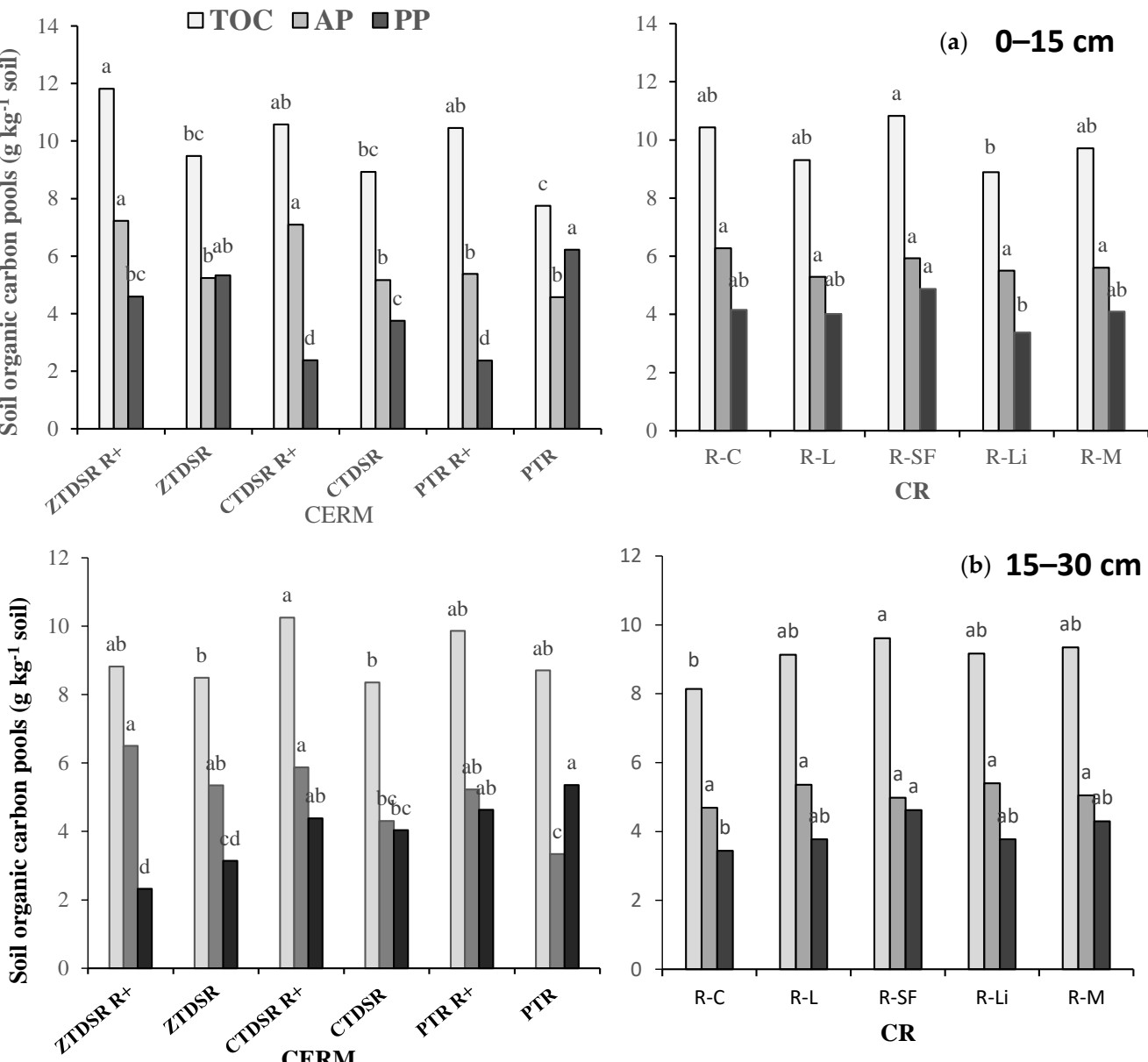

**Figure 4.** (**a**) 0–15 cm, (**b**) 15–30 cm. Active pool (AP), Passive pool (PP) (g kg$^{-1}$), and the TOC of bulk soil organic carbon. Means followed by the same letters do not differ significantly from each other by Tukey's HSD test at 5% probability level (*n* = 3).

*3.7. Carbon Management Index*

Tillage and residue-based crop establishment practices with different winter crop rotations strongly influenced the C-management indices (CMI) (Figure 5). The CMI in topsoil in ZTDSR$_{R+}$ was significantly increased by ~24% when compared to ZTDSR, though, a non-significant difference was observed between CTDSR$_{R+}$ and PTR$_{R+}$ production systems. Among the winter crop rotations, the higher value of CMI was observed under chickpeas and lentils but remained at par with others.

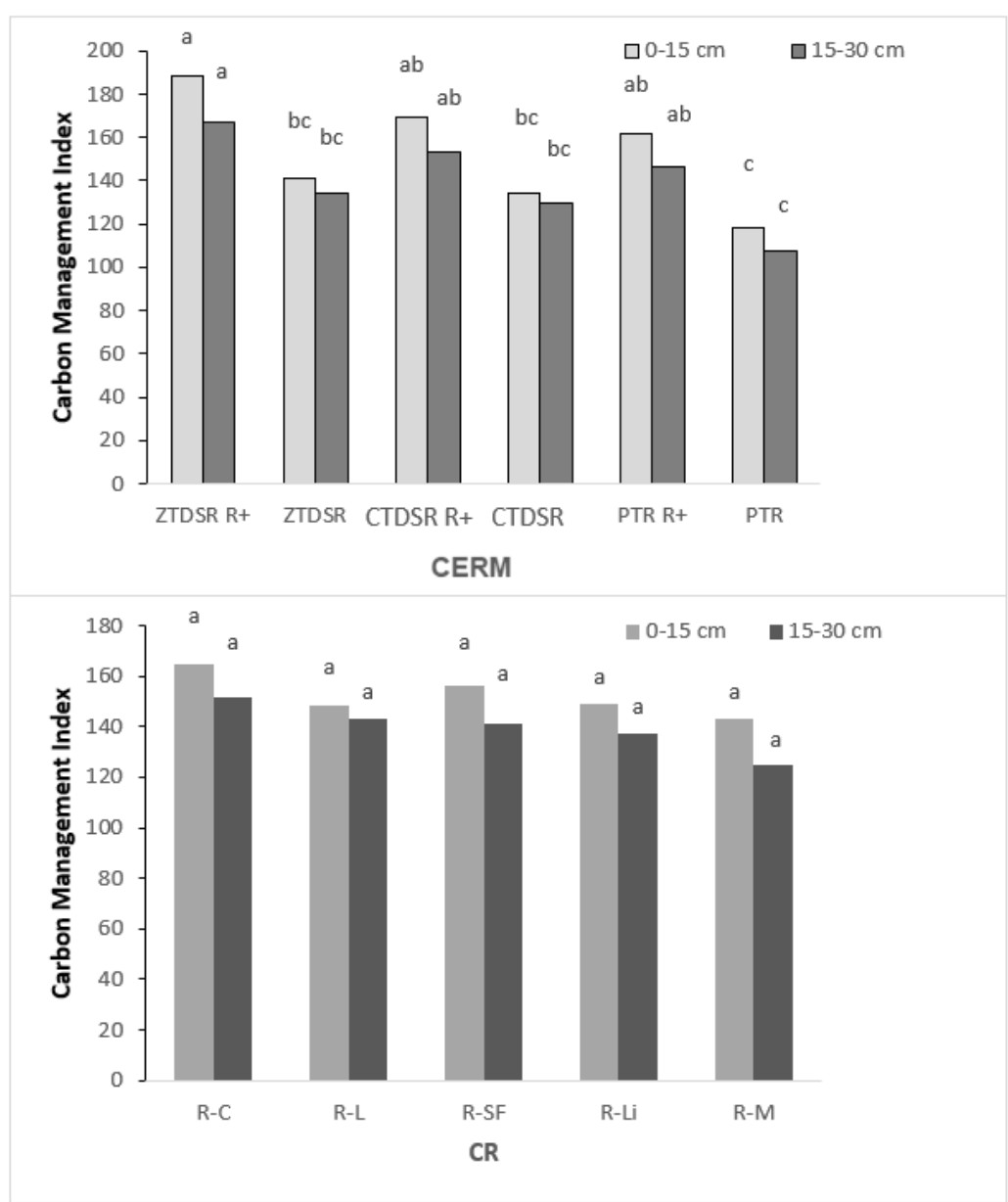

**Figure 5.** CMI as influenced by the CERM and CR (0–15 and 15–30 cm). ZTDSR: zero-tillage direct seeded rice; CTDSR: conventional till-direct seeded rice; PTR: puddle transplanted rice; R+: 30% residue retention, R-C: rice–chickpea; R-L: rice–lentil; R-SF: rice–safflower; R-Li: rice–linseed; R-M: rice–mustard. Means followed by the same letters do not differ significantly from each other by Tukey's HSD test at 5% probability level (*n* = 3).

### 3.8. Correlation between the Soil Properties

Pearson's correlation study of soil characteristics revealed a positive impact of CERM and CR activities (Figure 6). Overall, aggregate-associated C and SOC had significantly positive correlations ($p < 0.05$). Total water-stable aggregates (TWSA), water-stable macroaggregates (WSMacA), and SOC had significant positive relationships. SOC significantly correlated with CMicAC, CMacAC, and MesoAC but negatively correlated with FD.

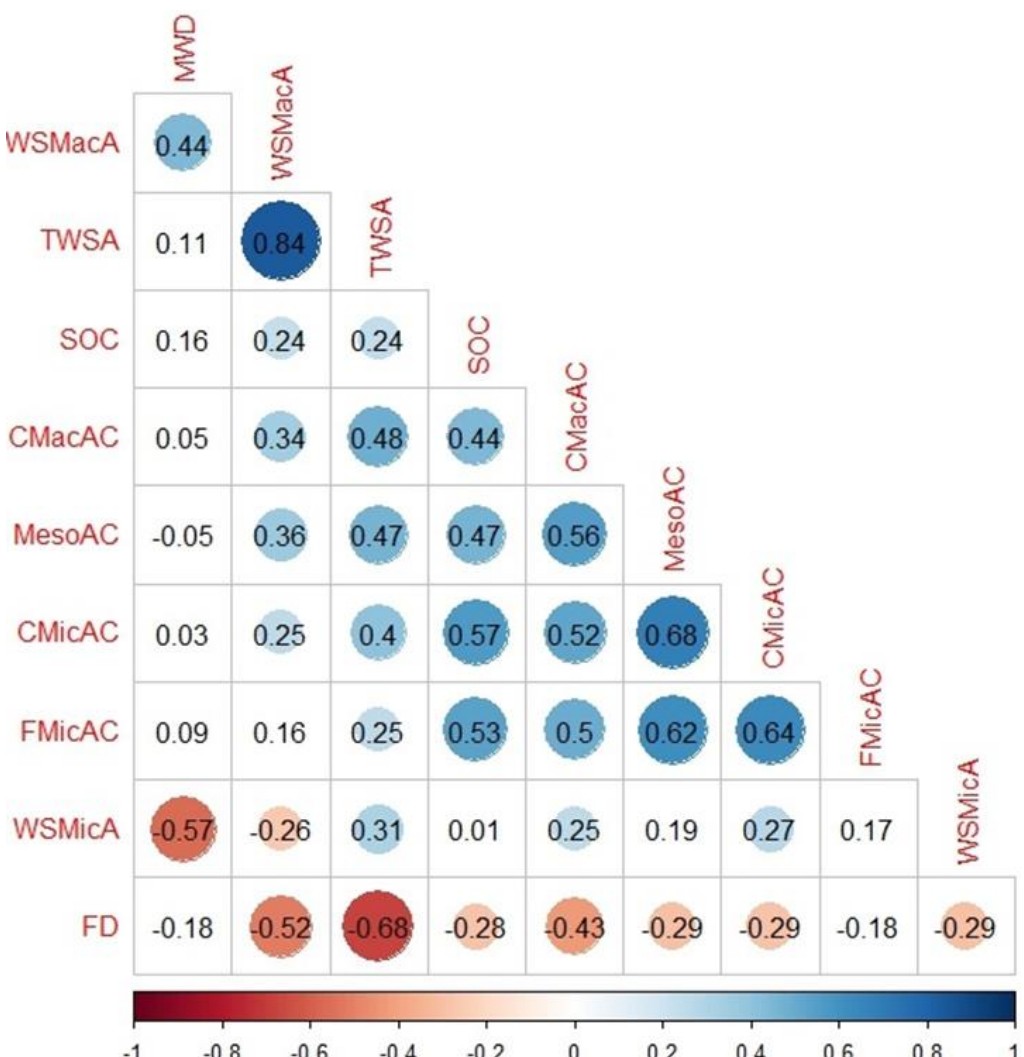

**Figure 6.** Correlation between soil properties as influenced by CERM and CR at the soil surface (0–15 cm). MWD: mean weight diameter, FD: fractal dimension, TWSA: total water stable aggregate, WSMacA: water stable macroaggregate, WSMicA: water stable microaggregate, CMacAC: coarse macroaggregate-associated C, MesoAC: meso-aggregate-associated C, CMicAC: coarse microaggregate-associated C, FMicAC: fine microaggregate-associated C. The correlation plot shows its significance (level 5%) with the shaded value with the other variables.

### 3.9. System Rice Equivalent Yield (SREY)/System Productivity

The maximum system productivity was observed in PTR$_{R+}$ (9.6 t ha$^{-1}$) in terms of rice equivalent yields (REY) but it remained at par with CTDSR$_{R+}$ (9.53 t ha$^{-1}$) and ZTDSR$_{R+}$ (9.31 t ha$^{-1}$), while ZTDSR (8.31 t ha$^{-1}$) and CTDSR (8.52 t ha$^{-1}$) produced lower productivity (Table 4). None of the yield differences among the different CERMs and CRs were statistically significant. Here, the study might draw an inference that conservation agriculture (CA)-based management practices (zero-tillage and residue retention) could produce comparable yields with better soil quality attributes in comparison to conventional management practices.

**Table 4.** Effect of CERM on system rice equivalent yield (SREY) under rice fallow conditions.

| CERM | System Rice Equivalent Yield (SREY) | | | | | Mean |
|---|---|---|---|---|---|---|
| | Chickpea | Lentil | Safflower | Linseed | Mustard | |
| $ZTDSR_{R+}$ | 10.18 ab | 10.11 a | 10.28 a | 6.65 bc | 9.32 abc | 9.31 |
| ZTDSR | 9.24 b | 9.12 a | 9.05 b | 5.78 d | 8.36 c | 8.31 |
| $CTDSR_{R+}$ | 10.53 a | 10.18 a | 9.85 ab | 7.35 ab | 9.74 ab | 9.53 |
| CTDSR | 9.41 ab | 9.14 a | 8.80 b | 6.49 cd | 8.76 bc | 8.52 |
| $PTR_{R+}$ | 10.11 ab | 10.11 a | 9.82 ab | 7.71 a | 10.25 a | 9.60 |
| PTR | 9.23 b | 9.26 a | 9.04 b | 7.17 abc | 9.39 abc | 8.82 |
| Mean | 9.78 | 9.65 | 9.47 | 6.86 | 9.30 | |

Means followed by the same letters in the columns do not differ significantly from each other by Tukey's HSD test at 5% probability level ($n = 3$). ZTDSR: zero-tillage direct seeded rice; CTDSR: conventional-till-direct seeded rice; PTR: puddle transplanted rice; R+: 30% residue retention.

## 4. Discussion

### 4.1. Water Stable Soil Aggregate

Tillage, crop rotation, and other soil management practices, in addition to the soil organic matter (SOM), play important roles in the formation of aggregates and in maintaining soil structure [27]. Less mechanical disturbance of the soil in ZTDSR and higher levels of organic material in residue-retained treatment resulted in an increased percentage of water-stable macro aggregates. Continuous tillage affects the soil's structural stability, speeds up microbial decomposition, and reduces the amount of organic matter or cementing agents in the soil [24]. When the soil organic matter is lost via constant plowing (TPR), a fraction of macroaggregates diminishes, resulting in an excess of microaggregates [28]. Rice–chickpea (R-C), rice–lentil (R-L), and rice–safflower (R-SF) systems enhanced the macroaggregates over other treatments, such as increased rhizospheric activity and root exudates of pulse crops. Thus, the $ZTDSR_{R+}$ production system and oilseed/pulse cropping rotations may work together to create water-stable macroaggregates.

The capacity of the soil aggregates to maintain stability when exposed to changing conditions was determined by their level of stability [29]. Increased SOM buildup of macro-aggregates may be the cause of higher MWD in CA operations, which indicates that there were bigger aggregates as evident from the wet sieving method [28]. CA-based treatments with higher proportions of macroaggregates enhanced C-sequestration and nutrient availability by providing appropriate aeration and water circulation in the root zone [28]. The GMD could influence the soil porosity and identify the prevalent aggregate size class in a soil sample. Higher GMD was mostly due to increased organic C-storage at the surface soil and a decrease in soil mobility in C-based management practices [30]. According to our findings, when soil depth increased, values of GMD/MWD decreased. When compared to CT treatments, an unstable aggregate index ($E_{LT}$) of CA-based management practices exhibited an opposite pattern. In our research, the $ZTDSR_{R+}$ production system significantly controlled soil erodibility by maintaining the crop residue and no-tillage operations on the surface of the soil.

### 4.2. Aggregate Fractal Dimension

$ZTDSR_{R+}$ treatment had the lowest D value at a 0–15 cm depth, owing to the retentions of crop residues in fields with less tillage that promote aggregation formation [31] (Figure 2). In the soil, depth crop rotation treatments had non-significant influences on the fractal dimensions of soil aggregates. The D value also increased with increasing soil depth. A higher D value was found in the PTR production system as compared to the rest of the treatments after the fourth year of experimentation of both soil depths. In long-term experiments, D could respond rapidly to tillage treatments and can be used to determine soil aggregate stability. Our findings are comparable to those of Zheng et al. [21].

### 4.3. Water-Stable Aggregate-Associated Carbon

The SOC is indicative of good agricultural soil and is a key component of the soil's health and qualities [32]. The SOC quantity, soil fertility, and crop yield have significant impacts on the creation and stability of water-stable soil aggregate structures [21]. We found that the amount of TOC in 0–15 cm is higher than in sub-layers (Table 3). The ZTDSR$_{R+}$, safflower, and chickpea systems had higher C in macroaggregates at 0–15 cm in depth. Because of the lesser soil disturbances, much bigger macroaggregate-stored C-concentration advocates delayed macroaggregate disintegration rates [33]. This indicates that CA-based management practices (ZTDSR) have the potential to preserve significantly more soil C inside macroaggregates, protecting the SOC against oxidation [34]. Pulse-inclusive rotation systems boosted the C buildup in macroaggregates (>2.0, 2–0.5, and 0.25–0.125 mm), while oilseeds increased the soil C in meso-aggregates (2–0.5 and 0.25–0.125 mm), which is an excellent indication of the sequestration of carbon [5]. Soil particles were typically kept together by organic residues to form macroaggregates [35].

### 4.4. Carbon Preservation Capacity (CPC) in Various Aggregate Classes

The CPC findings revealed that ZTDSR$_{R+}$ production systems enhanced the C storage of macroaggregates at 0–15 cm. Meso-aggregate fractions held the most C and had the best C storage capacity, followed by coarse microaggregates, indicating that these aggregates perform important roles in C sequestration under the ZTDSR$_{R+}$ production system. Our results confirmed that the less mechanical disturbance of soil with crop residue retention boosted the CPCs of aggregates. Song et al. [24] also found that medium-sized aggregates have greater specific surface areas and more active sites, making meso-aggregates the principal carrier of organic carbon. They can adsorb organic molecules because of enhanced ligand exchange and multivalent cation bridges.

### 4.5. Carbon Pools

In ZTDSR$_{R+}$ production systems, less exposure to external factors regulates reduced organic matter breakdown, resulting in passive C pool buildup. Crop residue on the surface likely created a hindrance to sunlight and the flow of air, slowing the breakdown process [14]. Buried crop residue in the PTR$_{R+}$ treatment degraded faster due to increased contact between soil and residue [36]. In this study, pulse-based cropping systems maintained better soil C while halting the loss of readily oxidizable C [37]. Growing leguminous crops for two years in a sub-tropical region previously resulted in more SOC content in poor soil [38]. As a consequence, it is clear that when employing pulse cropping sequences, ZTDSR$_{R+}$ was the most successful treatment for raising both LC and NLC carbon pools in soil. The addition of oilseed crops, such as safflower, also increased active pool carbons at the surface soil, and at 15–30 cm depths, TOC and passive pool carbons were raised. The plant roots collect and transmit C dioxide from the atmosphere into the soil as C-bearing compounds, which they store for longer times, such as AP and PP in soil [37]. In this way, CA-based management methods combined with pulses may have significant beneficial influences on soil health, especially at the SOC level.

### 4.6. Carbon Management Index (CMI)

The CMI of soil was significantly high in ZTDSR$_{R+}$ (46.3%) compared to those in the PTR production system in the top 0–15 cm of soil (Figure 5). A higher CMI was observed in oilseed (safflower) and pulse-based cropping systems (rice–chickpea) in both soil layers. CMI higher than 100 signifies a sustainable management system that maintains good soil quality [38]. It was noted that the PTR production system had lower rates of soil CMI. Overall, results indicated that incorporating oilseed (safflower) and pulses (chickpea/lentil) in cropping sequences based on rice enhanced the CMI. The CA-based management practices with the inclusion of pulses are noted to improve the SOC/CMI in different regions of India [39,40].

*4.7. System Rice Equivalent Yield (SREY)/System Productivity*

System productivity in a CA-based production system was comparable to the PTR system due to better productivity of the fallow crops in the CA-based management system. The rice yield was much higher in the puddled transplanted production system (PTR) as compared to the ZTDSR system due to wet puddling and flooding, which resulted in improved weed management and nutrient availability for the crops [41]. Contrary to this, severe weed problems in the ZT/CTDSR production system in the absence of puddling, alternate wetting, and drying favored more weed growth and soil sickness resulting in lower crop yields [19]. However, crop yields of subsequent oilseeds and pulses were higher when grown after ZTDSR compared to the PTR production systems due to the improvements in the physicochemical and biological properties of the soil [14]. Thus, yield loss in rice in the ZTDSR production system was compensated by gains in crop productivity of all subsequent fallow crops, resulting in an improvement of overall system productivity. Thus, the diversification and intensification with the inclusion of potential oilseeds and pulse crops (safflower, chickpea, lentil), along with the appropriate improved CERM management practices during the rainy crops may represent a better step toward improving the overall system productivity and soil resilience in the rice-fallow system of eastern India and similar agroecotypes around the globe.

**5. Conclusions**

From the present study, it can be concluded that diversification and intensification of rice-fallow systems with the inclusion of short-duration high-yielding oilseeds and pulse crops are viable options for the horizontal expansion of areas under oilseeds/pulses, as well as for the improvement of the overall system productivity and soil resilience in eastern India. In a rice-fallow system, CSA-based management practices significantly improved the soil aggregation, aggregate stability, and SOC content. $ZTDSR_{R+}$ combined with the addition of oilseed and pulse crops, as well as residue retention, had a positive impact on the overall health status of the soil. Thus, the present study concludes with the following points:

(1) Among the crop establishment methods with residues, the $ZTDSR_{R+}$ production system boosted the overall system productivity by 5.6%, water-stable macroaggregates by 60.1%, and CMI by 58% over puddled transplanted rice (PTR) at 0–15 cm soil depths.

(2) Among crop rotations with various winter/post-rainy crops, the inclusion of chickpeas and safflowers in rice-based production systems increased the overall system productivities by 5.2% and 1.8%, and CMIs by 14.1% and 6.3%, respectively, in comparison to the mustard-based production system at 0–15 cm.

Hence, CSA management practices, i.e., diversification and intensification of unutilized land of the rice-fallows system with suitable, potentially high-yielding oilseeds and pulses using residual soil moisture can help in sequestering more C, maintaining better soil health, and improving the overall system productivity of eastern India and similar agroecotypes across the globe.

**Supplementary Materials:** The following supporting information can be downloaded at: https://www.mdpi.com/article/10.3390/su141711056/s1, Figure S1: Mean monthly meteorological observations during 2018 and 2019. Table S1: Treatment description of the present experiment.

**Author Contributions:** Conceptualization, J.S.M. and A.U.; Data curation, R.K., A.K.S., S.M., R.S.M., J.S.C., M.K., H.S.R., N.R.S., S.K.Y., H.H., P.J., P.K.S. and R.K.R.; Formal analysis, A.K.S., S.M., H.S.R., N.R.S., S.K.Y., H.H., P.J., P.K.S. and R.K.R.; Investigation, K.S., J.S.M. and A.K.B.; Methodology, R.K. and S.M.; Project administration, R.K., J.S.M., A.K.B. and A.U.; Resources, A.K.S., J.S.C. and A.K.B.; Software, S.M.; Visualization, R.S.M. and A.U.; Writing—original draft, K.S.; Writing—review & editing, A.K.S. All authors have read and agreed to the published version of the manuscript.

**Funding:** We have received the financial and technical assistance via the ongoing project "Consortia Research Platform on Conservation Agriculture (CRP on CA)" from the ICAR-Indian Institute of Soil Science, Bhopal, Madhya Pradesh, India.

**Institutional Review Board Statement:** Not applicable.

**Informed Consent Statement:** Not applicable.

**Data Availability Statement:** Recorded data were statistically analyzed and are presented in the manuscript.

**Acknowledgments:** The authors sincerely acknowledge the contributions of three anonymous reviewers for their constructive and thoughtful comments to improve the quality of the manuscript. financial and technical assistance via the ongoing project "Consortia Research Platform on Conservation Agriculture (CRP on CA)" from the ICAR-Indian Institute of Soil Science, Bhopal, Madhya Pradesh, India.

**Conflicts of Interest:** The authors declare no conflict of interest.

## Abbreviations

| | |
|---|---|
| AP | active carbon pool |
| BD | bulk density |
| C | carbon |
| CA | conservation agriculture |
| CSA | climate-smart agriculture |
| CERM | crop establishment-cum-residue management |
| CMacAC | coarse macroaggregate-associated C |
| CMI | carbon management index |
| CMicAC | coarse microaggregate- associated C |
| CPC | carbon preservation capacity |
| CR | cropping rotation |
| CTDSR | conventional-till direct seeded rice |
| $CTDSR_{R+}$ | CTDSR with rice residue retention |
| DTPA | diethylenetriaminepentaacetic acid |
| FD | fractal dimension |
| FMicAC | fine microaggregate-associated C |
| GMD | geometric mean diameter |
| LC | labile carbon |
| LI | lability index |
| LLC | less labile carbon |
| M ha | million hectares |
| MesoAC | meso-aggregate-associated C |
| MWD | mean weight diameter |
| NLC | non-labile carbon: NLC |
| OC | organic carbon |
| P | phosphorus |
| PP | passive carbon pool |
| PTR | puddled transplanted rice |
| $PTR_{R+}$ | PTR with rice residue retention |
| R-C | rice–chickpea |
| REY | rice equivalent yield |
| R-L | rice–lentil |
| R-Li | rice–linseed |
| R-M | rice–mustard |
| R-SF | rice–safflower |
| SOC | soil organic carbon |
| SREY | system rice equivalent yield |
| TOC | total organic carbon |
| TWSA | total water-stable aggregate |
| VLC | very labile carbon: VLC |
| WSMacA | water stable macroaggregate |
| WSMicA | water stable microaggregate |
| ZTDSR | zero-tillage direct-seeded rice |
| $ZTDSR_{R+}$ | ZTDSR with rice residue retention |

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
