# Peer review of "Sustainable Intensification of Rice Fallows with Oilseeds and Pulses: Effects on Soil Aggregation, Organic Carbon Dynamics, and Crop Productivity in Eastern Indo-Gangetic Plains"

_sustainability, doi:10.3390/su141711056_

Round 1

Reviewer 1 Report

The analysis of variance table should be reported so readers can convince themselves that the split-plot nature of the experiment was properly taken into account. I am suspecting it was not. Most importantly, the test of the interaction should be provided. The linear mixed model used for analysis should be stated explicitly. Depending on the outcome of the test for interactions, the choice of mean comparisons should be made. Marginal means. The DMRT is not appropriate for error control. If the family-wise type I error rate is to be controlled, the Tukey test can be used. Otherwise, an LSD test should be used.

Reviewer 2 Report

Dear Editor

I carefully reviewed the manuscript entitled “Sustainable intensification of rice-fallows with oilseeds and pulses: effects on soil aggregation, organic carbon dynamics and crop productivity in eastern Indo-Gangetic Plains” (reference sustainability-1852275), by Saurabh et al., which submitted to Sustainability. The manuscript topic is adequate for the aims and scope of the journal.  The authors should address the following comments and revise the manuscript accordingly:

1. What is the novelty of the research in the international context? The necessity and innovation of the article should be presented in the last paragraph of introduction section.

2. The introduction section needs to be updated with the recent articles available for similar studies.

https://doi.org/10.1007/978-981-15-2985-6_8

3. In general, legends of figures are not all self-explainable. I am recommending that figures must be self-explanatory. That is, all statistics and abbreviations used must be clearly explained. The quality for all Figs in MS should be improved; I suggest to remade for the resolution. I have highly recommended the authors to introduce the mean comparison letters and/or standard error/deviation in the figures and tables.

4. How many replicates have been done for each assay. Please indicate it in the related sections.

5. Referrals in the text and references list should be in accordance with the journal's format.

6. Please provide the brands of chemicals used during the experiments throughout the materials and methods section.

8. There are some typographical errors in the paper and authors should be corrected them in the revised version. You should check the writing forms of the units in the whole text.

Reviewer 3 Report

Abstract:

Please change “was to identifying” to “was to identify”

Please change “had 58% more” to “was 58% more”

Please change “In PTR, system” to “In PTR system,”

Introduction:

The transition between the lines “India contributes 79% (11.7 M ha) of South Asia's total rice-fallows (15 m ha) (Kumar et al., 2020). Although India is approaching towards self sufficiency in pulses, the import of oilseeds is still a major concern.” is very abrupt. Please restructure the statements.

Please cite the data mentioned (such as oil recovery rates, productivity of oilseed crops)

“thr” to “the”

Please change” has a adverse impacts” to “has an adverse impact”

Please restructure the line” In winter, oilseeds i.e. mustard, linseed and safflowerand pulses i.e. chickpea, lentil, lathyrus, and field pea might be economically produced in rice-fallows to attain the self-sufficiency in oilseeds and pulses. short-duration Furthermore, pulse crops enhance soil health restoration by fixing nitrogen (N) from atmosphere (Kumar et al., 2019a).”

The transition between paragraphs is incoherent, please restructure

Please mention why you selected the treatments or what was the logic behind them. Which of the selected treatments are climate smart and why?

Materials and methods

How can we conclude a field study based on one year data.

2.2. Field management and experimental design:

Please mention the months of rainy season

2.3. Analysis of soil samples

Was soil analysis also done in 2016?

What was the reason for waiting 4 years before taking soil samples? how was the soil managed in those 4 years?

2.6. Statistical analysis

What were the fixed and random effects?

Based on values in Table 1, the distribution of data doesn’t seem normal for all treatments, what transformations were used to normalize data distribution

Results

Table 4: why mean is also included in mean separation? I  not why are there alphabets on means?

Discussion:

Concluding anything from the results is a stretch, all conclusions provided are vague. Please revise the study.

Also please consider adding line numbers to manuscript sent for review.

Round 2

Reviewer 1 Report

The ANOVA has all terms required but the analysis does not account for the fact that the mainplot error term rep*mp is a random effect. The authors need to use a mixed model package to get the correct analysis. This needs to include a method for approximating the denominator degrees of freedom.

The layout for the one replicate provided by the authors is not randomized. If this was indeed the layout, then the design was flawed. I need to see the layout of all three replicates as they were placed in the field trial to verify what was done.

Author Response

File attached

Reviewer 3 Report

Comments and suggestions have been adequately responded to. 

Author Response

Thank you very much for your kind suggestion for improving our manuscript significantly and considering/accepting our work for possible publication in the esteemed journal.

Round 3

Reviewer 1 Report

From the field layout and the responses given, it is now clear that the design was not a standard split-plot design as previously described, as will be detailed below, and that it was not properly randomized.

The main text says that there were six main plot treatments, but there are only three main plots. The treatment factor allocated to these main plots is tillage and crop establishment method with three levels (ZTDSR, CT-DSR and PTR). These were apparently allocated according to a Latin square with three super-rows and three super-columns. There is a second factor with two levels, i.e. with and without residue retention. This second factor, residue retention, is not allocated to main plots, however, but to rows within mainplots. The five crops (third factor) are allocated to five columns within main plots. Thus, the design for the factors residue retention and crop is a strip-plot design.

All of this would be fine with proper randomization. But as the authors reveal, proper randomization was not exercised. The authors argue that this is an experiment within a larger project and that for them the layout was a given, so nothing can be done at this stage to make up for the lack of randomization. This is true, but it does not make the problem go away. In my view, the only way this paper can be published is that the details about the layout as reported in the cover letter are fully included in the paper and a proper description of the layout along the lines sketched above is given as well, including the figure with the layout.

As regards analysis, the problem with lack of randomization is that strictly speaking no valid analysis can be provided. My suggestion is to fully reveal the lack of randomization, describe the layout as sketched above, and then perform an analysis under the model that would have been appropriate under proper randomization. That model is a three-factor model for treatments with factors tillage & crop establishment method (3 levels), residue retention (2 levels) and crop (5 levels). The design needs to be represented by adding these effects: super-rows (fixed), super-columns (fixed), main plot (random, 9 levels), row within main plot (random), column within main plot (random). The corresponding five design factors need to be properly coded, in addition to the three treatment factors. If such an analysis is furnished, I would like to see the full data in an Excel file and the SAS code used to produce the analysis.

In the current version, the authors give a brief description of their mixed model. In light of the above, that model is not sufficient. Also, tillage cannot be a random effect, it is a fixed factor. Main plot is a random factor, and a separate variable should be included in the dataset to model this properly. See this review for details:

Piepho, H.P., Büchse, A., Emrich, K. (2003): A hitchhiker's guide to the mixed model analysis of randomized experiments. Journal of Agronomy and Crop Science 189, 310-322.